# Competition-Exclusion for Manganese Is Involved in Antifungal Activity of Two Lactic Acid Bacteria Against Various Dairy Spoilage Fungi

**DOI:** 10.3390/microorganisms13112543

**Published:** 2025-11-06

**Authors:** Charlène Boulet, Emmanuel Coton, Marie-Laure Rouget, Florence Valence, Jérôme Mounier

**Affiliations:** 1Univ Brest, INRAE, Laboratoire Universitaire de Biodiversité et Ecologie Microbienne, F-29280 Plouzané, France; charlene.boulet@univ-brest.fr (C.B.); emmanuel.coton@univ-brest.fr (E.C.); 2Univ Brest, CNRS, UMS 3113, Institut Universitaire Européen de la Mer (IUEM), F-29280 Plouzané, France; marie-laure.rouget@univ-brest.fr; 3STLO, Institut Agro, INRAE, F-35000 Rennes, France; florence.valence-bertel@inrae.fr

**Keywords:** bioprotection, food spoilage, *Lactiplantibacillus plantarum*, *Lactobacillus rhamnosus*

## Abstract

Biopreservation using lactic acid bacteria has gained a growing interest as an alternative to chemical preservatives and/or as a complementary tool to prevent fungal spoilage in dairy products. Among the action mechanisms of antifungal LAB, competitionexclusion for trace elements has recently been highlighted. To further investigate this mechanism, two antifungal LAB strains, *Lactiplantibacillus plantarum* L244 and *Lactobacillus rhamnosus* CIRM-BIA1759, were studied in a yogurt model. Firstly, the antifungal activity of these strains against four main dairy spoilage fungi (*Penicillium biforme*, *Mucor racemosus*, *Galactomyces geotrichum* and *Yarrowia lipolytica*) was evaluated with or without trace element (6 metals and 12 vitamins) supplementation. Only manganese supplementation led to a suppression of the antifungal activity of both *L. plantarum* L244 and *L. rhamnosus* CIRM-BIA1759 against *P. biforme* and/or *Y. lipolytica*. The scavenging of trace elements was then measured using HR-ICP-MS in both cell-free yogurt whey and fungal biomass. HR-ICP-MS results showed a significant scavenging of Mn in *L. plantarum* L244 and *L. rhamnosus* CIRM-BIA1759 whey, as well as Cu for *L. rhamnosus* CIRM-BIA1759. Moreover, element uptake profiles, including metal and non-metal elements, for each of the target fungi were affected by the use of antifungal cultures. Finally, the role of competitionexclusion for manganese in the inhibition of 25 fungal spoilers was evaluated via oCelloScope growth follow-up. Growth inhibition by antifungal LAB strains was suppressed after Mn supplementation in cell-free whey for the 16 (out of 25) fungi initially inhibited without Mn supplementation. The nine other fungi were not inhibited or were poorly inhibited in the different tested conditions. This study confirmed the role of competitionexclusion for Mn in the antifungal activity of *L. plantarum* L244 and *L. rhamnosus* CIRM-BIA1759 strains but also revealed that this mechanism is not generic among fungal species, as the growth behavior of several tested species was not impacted by Mn scavenging.

## 1. Introduction

Dairy products are susceptible to acid-tolerant fungal contamination (yeasts and molds), the growth of which can lead to spoilage and thus food losses and waste [1,2,3,4]. In order to control fungal spoilage in dairy products, multiple hurdle technologies, including prevention and control methods, are combined [4]. Control methods can be divided into inactivation methods (e.g., heat treatment) and methods which can slow down or inhibit fungal growth during storage, such as the use of antifungal preservatives comprising weak organic acids (e.g., potassium sorbate) and natamycin, a polyethylene antibiotic. However, fungal resistance or adaptation to these antifungal molecules [5] and, more importantly, the consumer’s demand for less-heavily processed and preservative-free products have led to the development of more “natural” solutions [6], such as the use of antifungal bioprotective cultures. Among antifungal bioprotective agents, lactic acid bacteria (LAB) have generated a growing interest as an alternative to chemical preservatives [7,8] due to their long history of safe use in a variety of fermented foods and their antifungal activity potential [9,10]. Furthermore, many of them, including both *Lactiplantibacillus plantarum* and *Lactobacillus rhamnosus* species [11,12], have the QPS (Qualified Presumption of Safety) status. Therefore, biopreservation with antifungal LAB and/or their metabolites, used to prevent or delay fungal spoilage, has become one of the emerging strategies. However, antifungal LAB modes of action are still not fully known from a mechanistic point of view. Until recently, the production of antifungal compounds coupled with a pH decrease was the main described mechanism involved in LAB antifungal activity [10]. These antifungal metabolites are produced through central carbon metabolism, secondary metabolism or bioconversion. They include organic acids, fatty acids, cyclopeptides and volatile compounds, to name just a few [10]. They have different impacts on cell functions: cell wall instability/permeability, proton gradient interference, oxidative stress and enzyme inhibition. However, the additive and/or synergistic effects of these metabolites have been neglected, with a main focus on only individual compound effects. Indeed, it is still difficult to quantify their effect because the concentration at which they are observed in biopreserved dairy products is far below their determined minimal inhibitory concentration (MIC) values [13]. It is therefore likely that other mechanisms are involved in antifungal activity.

In 2020, Siedler et al. [14] demonstrated, using yogurt-derived cell-free whey, that among five trace metal elements (iron, copper, zinc, magnesium and manganese), competitive exclusion for manganese (Mn) was a major mechanism in two selected antifungal lactobacilli. This was primarily due to a particularly high expression of a mntH manganese transporter-encoding gene (*mntH1*), which reduced the Mn level available for fungal growth by active transport. The depletion of manganese as a bioprotective strategy was used as the base of a patent entitled “Inhibition of fungal growth by manganese depletion” (EP3780962A2). Mn is an essential trace element as a cofactor in all kingdoms of life. In *L. rhamnosus* and *L. plantarum*, manganese plays several rolesIt is involved in oxidative stress resistance through high intracellular accumulation and is used for the non-enzymatic detoxification of superoxide radicals [15]. Furthermore, in *L. plantarum*, it is incorporated into Mn-catalase for the removal of endogenous H_2_O_2_ under aerobic growth. In addition, manganese is also considered to be crucial for growth, since various enzymes related to carbon metabolism and ATP-generation, such as lactate dehydrogenase, are known to be Mn-dependent or highly stimulated by manganese. In fungi, Mn serves as a micronutrient for superoxide dismutases (i.e., mitochondrial Mn-SOD) and protein glycosyltransferases in the Golgi, but is also a cofactor of kinases and phosphatases [16]. Nevertheless, while it is clear that manganese scavenging is an important mechanism in antifungal lactobacilli, other trace elements which are limiting in dairy products, including other metals and vitamins, may also be involved in competitionexclusion phenomena and should thus be explored. In addition, it is not clear yet whether this action mechanism is generic among the large diversity of fungi able to spoil dairy products.

In this context, the aim of this study was to extend the reported Mn competition exclusion and investigate if competitive exclusion mechanism also concerned other trace elements and was applicable to a larger panel of fungal species. To do so, two known antifungal LAB strains, namely *L. plantarum* L244 and *L. rhamnosus* CIRM-BIA1759, were studied using a yogurt model [10,13]. First, the impact of the individual supplementation of 20 trace elements (metals, including manganese, and vitamins) on antifungal activity was studied. Then, to quantify trace metal scavenging, HR-ICP-MS (High-Resolution Inductively Coupled Plasma Mass Spectrometry) was performed on cell-free whey obtained from yogurts inoculated or not inoculated with the antifungal LAB strains, and on fungal biomass grown in these wheys. Finally, the potential involvement of competitionexclusion for manganese on the antifungal activity towards 25 fungal species was studied via oCelloScope growth kinetics.

## 2. Materials and Methods

### 2.1. Microorganisms and Culture Conditions

The antifungal LAB *L. plantarum* L244 and *L. rhamnosus* CIRM-BIA1759 were obtained from the Laboratoire Universitaire de Biodiversité et Ecologie Microbienne collection (LUBEM, France) [17] and CIRM-BIA culture collection (INRAE, France, https://collection-cirmbia.fr/, accessed on 27 October 2025) [18], respectively. These strains were stored at −80 °C in milliQ water supplemented with 20% glycerol. Both strains were cultivated twice in MRS (Mann–Rogosa–Sharpe) broth (Biomérieux, Marcy-l’Étoile, France) at 30 °C for 24 h. After centrifugation at 9000 rpm for 15 min of a sufficient volume of the pre-culture to reach the target cell concentration, cells were resuspended in sterile skimmed milk to reach 2 × 10^8^ CFU/mL. The LAB commercial starter culture used for yogurt production (MY800-Choozit, Danisco, France) was composed of *Streptococcus thermophilus*, *Lactobacillus delbrueckii* subsp. *lactis* and *L. delbrueckii* subsp. *bulgaricus*.

The 25 selected fungal targets, previously isolated from spoiled dairy products [19], included 11 yeasts and 14 molds, and were obtained from the Université de Bretagne Occidentale Culture Collection [17] and LUBEM collection. They corresponded to *Candida inconspicua* Euf6, *Candida intermedia* UBOCC-A-217012, *Candida parapsilosis* UBOCC-A-216002, *Cladosporium sphaerospermum* UBOCC-A-113031, *Cutaneotrichosporon suis* UBOCC-A-218003, *Debaryomyces hansenii* UBOCC-A-217015, *Galactomyces geotrichum* UBOCC-A-216001, *Kluyveromyces lactis* UBOCC-A-217011, *Meyerozyma guilliermondii* UBOCC-A-216003, *Mucor circinelloides* UBOCC-A-109182, *Mucor racemosus* UBOCC-A-116002, *Penicillium adametzoides* Ai10, *Penicillium antarticum* UBOCC-A-117326, *Penicillium bialowiezense* UBOCC-A-117365, *Penicillium biforme* UBOCC-A-116003, *Penicillium charlesii* Ai7b, *Penicillium roqueforti* Ef25, Penicillium solitum Ay2, *Phoma pinodella* UBOCC-A-116004, *Pichia fermentans* Euf17, *Rhodotorula mucilaginosa* UBOCC-A-216004, *Scopulariopsis candida* UBOCC-A-108117, *Thamnidium elegans* UBOCC-A-108122, *Trichosporon asahii* UBOCC-A-216005 and *Yarrowia lipolytica* UBOCC-A-216006. Yeast cell and mold spore suspensions were prepared as previously described [20]. Prior to antifungal activity testing, these suspensions were diluted using milliQ water to reach 5 × 10^3^ spores or cells/mL.

### 2.2. Impact of Trace Element Supplementation on LAB Antifungal Activity

The high-throughput method developed by Garnier et al. [21] was used to assess the antifungal activity on the yogurt matrix. Yogurt was prepared following the protocol developed by Leyva Salas et al. [22] with modifications. A mix of skimmed milk (Carrefour, France) and skimmed milk powder at 4% (Régilait, France) was firstly prepared. Then, a heat treatment of 30 min at 85 °C was applied before quick cooling of milk at 45 °C, and 0.1 g/L of the lyophilized starter culture (Choozit-MY800, Danisco, France) and a pH indicator (Litmus dye) were added. The obtained milk preparation was distributed into 24-well plates (2 mL/well) after homogenization. Antifungal strain suspensions prepared as described above were inoculated at 10^7^ CFU/mL. The tested metal and vitamin solutions were individually added at 10 times the concentration reported by the Ciqual values for yogurt (ANSES, 2020) [23] and data from Staggs et al. [24] and Zekai and Dağ [25]. Six metals, namely manganese (MnSO_4_ at 2.9 µg/g), copper (CuSO_4_ at 1.6 µg/g), iron (FeSO_4_ at 14.4 µg/g), zinc (ZnSO_4_ at 34.2 µg/g), nickel (Ni(SO_3_NH_2_)_2_ at 0.4 µg/g) and cobalt (CoCl_2_ at 24.3 ng/g) and twelve vitamins, namely B1 (at 2.9 µg/g), B2 (at 20.7 µg/g), B3 (at 13.5 µg/g), B5 (at 31.5 µg/g), B6 (at 4.6 µg/g), B7 (at 7.6 ng/g), B9 (at 1.8 µg/g), B12 (at 21.6 ng/g), C (at 25.2 µg/g), D (at 43.2 ng/g), E (at 9 µg/g) and K1 (at 33.3 ng/g), were tested. Controls corresponded to the absence of trace element supplementation. Plates were then incubated at 42 °C during 4 h with pH measurements taken every hour until pH 5 was reached. The syneresis liquid was removed at the end of fermentation and the obtained mini-yogurts were spotted in the center with 50 spores or cells of the target fungi (*M. racemosus* UBOCC-A-116002, *P. biforme* UBOCC-A-116003, *Y. lipolytica* UBOCC-A-216006 or *G. geotrichum* UBOCC-A-216001; one tested fungus per plate). Antifungal activity was assessed daily by visually evaluating fungal growth in comparison with the negative controls (without antifungal LAB, and supplemented or not supplemented with trace elements), after incubation for up to 20 days at 10 °C. Antifungal activity scoring was measured using the qualitative system developed by Garnier et al. [21] (i.e., − for no inhibition, + for slight inhibition, ++ for intermediate inhibition and +++ for total inhibition).

### 2.3. Scavenging of Yogurt Trace Elements by Antifungal LAB

Quantification of trace element scavenging by LAB in yogurts was determined using an HR-ICP-MS (Element XR, Thermo Scientific, Waltham, MA, USA) at the Pôle Spectrométrie Océan (IUEM, Plouzané, France). Samples containing 10 mL of cell-free whey were obtained through the centrifugation at 9000 rpm for 15 min of 20 mL yogurts (control yogurts and yogurts made with antifungal LAB at 10^7^ CFU/mL) stored for 2 weeks at 10 °C (four replicates per condition). Then, calcium, cobalt, copper, iron, magnesium, manganese, molybdenum, nickel, phosphorus, selenium, sodium, sulfur and zinc were quantified. To do so, for the extraction step, 1 g of whey was collected, digested overnight in PTFE vessels (Milestone MLS, Sorisole BG, Italy) with 1.5 mL of 30% H_2_O_2_ (Sigma, Saint-Quentin Fallavier, France) and 7 mL of concentrated 65% HNO_3_ (Sigma, Saint-Quentin Fallavier, France) and then placed in an ETHOS One high-performance microwave digestion system (Milestone MLS, Sorisole, BG, Italy) for 2 × 15 min at 200 °C. After cooling, the samples were completely evaporated at 90 °C. Then, cell pellets were resuspended in 20 mL of MilliQ water (Millipore, Feltham, UK) with 3.5% HNO_3_ and analyzed using the HR-ICP-MS in medium (MR = 4000) and high resolution (HR = 10,000) depending on the targeted isotope. HR-ICP-MS instrument operating conditions were the following: plasma RF power, 1200 W; cooling gas flow rate, 16 L/min; auxiliary gas flow rate, 0.9 L/min; sample gas flow rate, 1.08 L/min. All quantification standards were prepared from 1000 µg/mL PlasmaCal Single Element calibration standard solutions for ICP-AES and ICP-MS (SCP Science, Baie-D’Urfé, QC, Canada).

### 2.4. Uptake of Trace Elements by Fungi in Yogurt Cell-Free Whey

Uptake of trace elements was also determined for the first four selected fungal targets, *G. geotrichum* UBOCC-A-216001, *M. racemosus* UBOCC-A-116002, *P. biforme* UBOCC-A-116003 and *Y. lipolytica* UBOCC-A-216006, following the method described by Wehmeier et al. [26] with modifications. Yeasts and molds were inoculated at 10^3^ spores or cells/mL in a 100 mL Erlenmeyer flask containing 30 mL cell-free whey obtained from yogurts stored for 2 weeks at 10 °C, as described above, followed by incubation at 10°C for 1 week with shaking at 200 rpm (four replicates per condition). Cell pellets were then collected through centrifugation for 5 min at 9000 rpm, washed twice with MilliQ water and dried in a drying oven at 70 °C for 2 days; the biomass was finally determined using a precision balance through subtraction of the tube weight. For the extraction step, dried cell pellets were lysed overnight in PP-tubes (Corning, Sunderland, UK) using 1 mL of 65% HNO_3_ (Sigma, Saint-Quentin Fallavier, France) and 2 mL of 30% H_2_O_2_ (Sigma, Saint-Quentin Fallavier, France). Then, 3 mL of 65% HNO_3_ and 6 mL of 30% H_2_O_2_ were added, and extracted samples were poured in PTFE vessels before a microwave cycle of 5 min at 50 °C, 5 min at 75 °C and 30 min at 95 °C. The following steps were the same as previously described for whey, except that cell pellets were resuspended in a final volume of 10 mL milliQ water with 3.5% HNO_3_ before analysis.

#### 2.4.1. Growth Kinetics of Fungal Targets in Yogurt Cell-Free Whey

Growth kinetics at 10 °C of the first 4 targeted fungal species and 21 additional ones were determined in cell-free whey obtained as described above, supplemented or not supplemented with 2.9 µg/g MnSO_4_ after fermentation. The selected fungal targets were inoculated at 10^3^ spores or cells/mL, and the solution was distributed in 96-well plates (200 µL/well). Fungal growth kinetics were followed for 7 days at 10 °C using an oCelloScope (BioSense Solutions ApS, Farum, Denmark) with three replicates per tested condition. Modeling of growth parameters (µmax, i.e., maximum growth rate and lag, i.e., duration of lag time) was implemented using the QurvE software [27]. Then, in order to quantify and compare among fungal species the extent to which Mn scavenging by antifungal LAB was involved in their antifungal activity, two quantifiers were calculated, namely relative lag increase and relative µmax decrease, for each tested fungi and tested condition, as follows:(1)$$\mathrm{Relative}\;\mathrm{lag}\;\mathrm{increase}\;(\%)\;=\;(1-\frac{\left(Calculated\;mean\;lag\;value\;of\;control\right)}{\left(Calculated\;mean\;lag\;value\;of\;tested\;condition\right)})\times 100$$
(2)$$\mathrm{Relative}\;µ\max\;\mathrm{decrease}\;(\%)=(1-\frac{\left(Calculated\;mean\;µmax\;value\;of\;tested\;condition\right)}{\left(Calculated\;mean\;µmax\;value\;of\;control\right)})\times 100$$
where the term “tested condition” corresponded to whey obtained with the use of yogurt starters and either L244 or CIRM-BIA1759 antifungal LAB, and either with or without manganese supplementation. The term “control” corresponded to whey obtained using only yogurt starter cultures. Negative values were set to 0.

#### 2.4.2. Statistical Analyses

Statistical analyses were performed using R (R version 4.4.2, RStudio environment). The Kruskall–Wallis test or One-way analysis of variance (ANOVA) was carried out to detect significant differences between element concentrations in the different tested conditions. When significant differences (*p* < 0.05) were obtained, post hoc Dunn or Tukey tests were applied for pair comparison.

A principal component analysis (PCA) was also performed to visualize element uptake by fungi using the FactoMineR and Factoextra packages, and plots were generated using the ggplot2 package. A Student *t*-test was applied to compare element uptake expressed in µg or ng/g dry fungal biomass after cultivation in cell-free whey obtained with or without antifungal cultures. A heat map was generated using the *pheatmap* function to visualize the impact of manganese supplementation on the relative lag increase and µmax decrease in fungi cultivated in cell-free whey obtained with antifungal cultures and supplemented or not supplemented with Mn.

## 3. Results

### 3.1. Impact of Trace Element Supplementation on LAB Antifungal Activity

In the controls (i.e., yogurt produced with the MY800 starter culture but without antifungal bacteria), visible fungal growth was observed after 3 days for *M. racemosus* and *G. geotrichum*, 5 days for *P. biforme* and 6 days for *Y. lipolytica*. The tested conditions with the antifungal LAB but without trace element supplementation showed the antifungal activity of *L. plantarum* L244 against *P. biforme* and *Y. lipolytica*, while *L. rhamnosus* CIRM-BIA1759 only inhibited *P. biforme* (Figure 1). For *P. biforme*, while its growth was visible after 5 days for both antifungal LAB conditions, an intermediate antifungal activity (see Materials and Methods) was observed during storage (up to 19 days) (Figure 1). For *Y. lipolytica*, a 1-day delay for the time to visible growth was observed in the presence of *L. plantarum* L244, which was associated with an intermediate antifungal activity at 19 days (Figure 1). No visible inhibition of *M. racemosus* or *G. geotrichum* was observed with either antifungal LAB. After manganese addition at 10 times the concentration reported in yogurt, the observed fungal growth inhibitions by *L. plantarum* L244 and *L. rhamnosus* CIRM-BIA1759 (i.e., against *P. biforme* and *Y. lipolytica*, and only *P. biforme*, respectively) were suppressed, as the visual fungal growths were similar to those observed in the controls without the antifungal LAB strains (Figure 1). Regarding the additional tested metals (copper, iron, zinc, nickel and cobalt) and vitamins (B1, B2, B3, B5, B6, B7, B9, B12, C, D, E and K1) present in yogurt and also tested by supplementation, no effect on the visual growth or aspect of the four targeted fungi was observed (Appendix A).

### 3.2. Scavenging of Trace Elements in Yogurt by Antifungal LAB

To evaluate the trace element scavenging of the studied antifungal LAB strains, a quantification of trace elements was performed using HR-ICP-MS on cell-free whey from control yogurts (yogurt obtained only with MY800 starter cultures) and yogurts inoculated with either *L. plantarum* L244 or *L. rhamnosus* CIRM-BIA1759 and stored for 2 weeks at 10 °C. Significant decreases in Mn (*p* < 0.05) were observed for the whey from yogurts inoculated with *L. plantarum* L244 and *L. rhamnosus* CIRM-BIA1759 in comparison to the control (Figure 2A). Indeed, while Mn concentration was 7.98 ± 0.20 ng/g in the control; less than 1 ng/g was found in the yogurt whey produced with the antifungal LAB strains (0.54 ± 0.87 and 0.44 ± 0.38 ng/g for *L. plantarum* L244 and *L. rhamnosus* CIRM-BIA1759, respectively). A significant Cu decrease from 13.01 ± 0.32 to 7.88 ± 0.58 ng/g was also observed in the *L. rhamnosus* CIRM-BIA1759 yogurt whey as compared to the control (Figure 2B). The other trace elements, namely Na, Mg, P, S, Ca, Zn, Fe, Co, Ni, Mo and Se, did not show any significant concentration variations in the presence of antifungal LAB as compared to the control (Appendix A).

### 3.3. Uptake of Trace Elements by Fungi in Whey

In order to evaluate the scavenging of trace elements by the four initial fungal targets, dry biomass measurements were performed on the fungal cultures obtained after growth for 1 week at 10 °C and 200 rpm in either control, *L. plantarum* L244 or *L. rhamnosus* CIRM-BIA1759 whey, followed by HR-ICP-MS analyses.

As shown in Appendix A, the mean dry biomasses of *P. biforme*, *G. geotrichum* and *Y. lipolytica*, obtained after cultivation in *L. plantarum* L244 whey, were significantly lower (*p* < 0.05) than that from the control whey. The *M. racemosus* mean dry biomass, despite being lower, was not statistically different (*p* > 0.05) from the one obtained in the control whey (Appendix A). It is also worth mentioning that only two replicates of *P. biforme* culture in *L. plantarum* L244 whey yielded a sufficient biomass for dry biomass measurements and the subsequent element uptake analysis. The effect of *L. rhamnosus* CIRM-BIA1759 on fungal growth was less pronounced, with no significant differences in the mean dry biomass for *Y. lipolytica* and *M. racemosus*, and a significantly higher biomass for *G. geotrichum* as compared to the control. The only exception was *P. biforme*, which did not yield sufficient biomass in this condition for dry biomass measurements and trace element analysis.

The element uptake profile of each tested fungus in the different conditions is represented in Appendix A and was represented using a PCA (Figure 3A). The first axis (PC1) explained 36.2% of the total variance, while the second one (PC2) explained 21%. In the biplot of Figure 3A, individual variables with cos^2^ ≥ 0.25 and cell-free whey origin (noted as “no antifungal cultures”, “*L. plantarum* L244” or “*L. rhamnosus* CIRM-BIA1759”) are plotted, providing supplementary qualitative information. As shown, element uptake profiles, including metal and non-metal elements, by each of the tested fungi were affected by the use of antifungal cultures. Concerning metal elements, the biomass of fungal strains cultivated in yogurt whey prepared with antifungal cultures contained, as expected, significantly far less manganese (between 0.39-fold decrease for *P. biforme* and 0.008- to 0.03-fold decrease for the other tested fungal strains) as compared to the control conditions (Figure 3B and Appendix A). The Mg level and Mg and Cu levels were also slightly significantly lower (0.09- to 1.3-fold decrease) as compared to the control condition for *M. racemosus* in *L. plantarum* L244 whey, and in *G. geotrichum* biomass cultivated in *L. rhamnosus* CIRM-BIA1759 whey, respectively (Figure 3B and Appendix A). We also observed a significantly lower acquisition (0.2- to 0.9-fold decrease) of Ca and Na for *P. biforme*, *M. racemosus* and *G. geotrichum* after cultivation in *L. plantarum* L244 whey. In contrast, other metal elements were taken up by the tested fungal strains at significantly higher concentrations in the yogurt whey prepared with antifungal cultures. For example, *P. biforme*, *G. geotrichum* and *Y. lipolytica* acquired significantly more Zn, while *P. biforme* and *M. racemosus* acquired significantly more Fe (Figure 3B).

Concerning non-metal elements, P and S, and S alone, were significantly more abundant in *P. biforme* and *G. geotrichum* biomass obtained from *L. plantarum* L244 whey, respectively. Conversely, lower concentrations of P and S were observed in *G. geotrichum* biomass cultivated in *L. rhamnosus* CIRM-BIA1759 whey.

### 3.4. Growth Kinetics of Fungal Targets in Whey

In order to evaluate both the antifungal activity spectrum of *L. plantarum* L244 and *L. rhamnosus* CIRM-BIA1759 and the potential contribution of Mn to said antifungal activity, the growth kinetics of 25 fungal targets (yeasts and molds), including the 4 previously mentioned, were followed by oCelloScope in cell-free whey.

First, a qualitative assessment of the effect of manganese supplementation on fungal growth in cell-free antifungal whey, supplemented or not supplemented after fermentation with 2.9 µg/g MnSO_4_, was made by comparing oCelloScope images at a given time of incubation and for a given fungal strain (Figure 4).

Among yeasts, 7 out of the 12 tested, namely *C. parapsilosis, C. suis*, *D. hansenii*, *M. guilliermondii*, *R. mucilaginosa*, *T. asahii* and *Y. lipolytica*, were notably inhibited in *L. plantarum* L244 or *L. rhamnosus* CIRM-BIA1759 whey as compared to the control whey without antifungal strains (Figure 4A). In the conditions with manganese supplementation, the previous strains showed an equivalent growth as in the two control whey (i.e., without antifungal strains, and either with or without Mn supplementation). On the other hand, *C. inconspicua*, *C. intermedia*, *G. geotrichum*, *K. lactis* and *P. fermentans* were not sensitive to either *L. plantarum* L244 or *L. rhamnosus* CIRM-BIA1759 whey.

Among molds, 9 out of the 13 tested, namely *C. sphaerospermum*, *P. adametzoides*, *P. antarticum*, *P. bialowiezense*, *P. biforme*, *P. charlesii*, *P. solitum*, *P. pinodella* and *S. candida*, were notably inhibited in *L. plantarum* L244 or *L. rhamnosus* CIRM-BIA1759 whey as compared to the control whey after incubation for the same duration (Figure 4B). Interestingly, after manganese addition, the observed antifungal activity was suppressed. In contrast, the observed growths for *M. circinelloides*, *M. racemosus*, *P. roqueforti* and *T. elegans* were similar in either the *L. plantarum* L244 and *L. rhamnosus* CIRM-BIA1759 wheys or the control whey.

A comparison of oCelloScope images between conditions showed that cultivation in antifungal LAB whey had an impact on fungal cell morphology. Indeed, it was observed that *C. sphaerospermum*, *P. adametzoides*, *P. antarticum*, *P. bialowiezense*, *P. biforme*, *P. charlesii*, *P. solitum*, *P. pinodella* and *S. candida* hyphae were irregular and/or swollen (Figure 4B). Furthermore, for *C. sphaerospermum*, *P. antarticum*, *P. bialowiezense*, *P. solitum* and *P. pinodella*, spore swelling was observed, which was not the case in the control whey, and an impact on spore germination (no or delayed germination) was also observed (Figure 4B).

Based on the oCelloScope data, the lag time and µmax were determined for the 25 selected fungal targets in whey with or without the antifungal LAB, and supplemented or not supplemented with manganese. Independently of the considered whey, the lag time varied greatly depending on the tested fungi, with *T. asahii* showing the longest lag time and *T. elegans* the shortest (Appendix A), while *P. charlesii* and *P. adametzoides* showed the lowest µmax and *G. geotrichum* the highest (Appendix A).

In order to quantify and compare among fungal species the effect of antifungal cultures and the extent to which manganese scavenging by antifungal LAB was involved in their antifungal activity, the relative increase in lag time (Figure 5A) and relative decrease in growth rate were calculated and plotted in a heatmap (Figure 5B). Concerning the effect of antifungal cultures on the lag time of fungi (Figure 5A and Appendix A), the results showed that the lag times of 7 out the 25 tested fungi were impacted by both antifungal cultures, with a relative increase in lag time above ~20% as compared to the control condition. Increased lag times were observed for *P. antarticum* (47.81% and 48.33% increase in lag time when cultivated in *L. plantarum* L244 and *L. rhamnosus* CIRM-BIA1759 whey, respectively), *D. hansenii* (34.99% and 42.57%), *M. guilliermondii* (34.32% and 35.81%), *P. pinodella* (30.42% and 32.61%), *Y. lipolytica* (28.30% and 35.87%), *C. suis* (23.38% and 24.12%) and *P. bialowiezense* (19.53% and 22.21%). In addition, the lag times of *T. asahii*, *C. sphaerospermum*, *T. elegans*, *P. solitum* and *P. charlesii* were, respectively, increased by 33.84, 28.80, 28.10, 24.9 and 21.67% in *L. rhamnosus* CIRM-BIA1759 whey only. As shown in Figure 5A, Mn supplementation in antifungal whey totally reversed the effect observed on lag time for the species mentioned above, with the exception of *P. adametzoides*, *P. antarticum*, *P. charlesii* and *C. suis*, for which relative lag time increases were still above 20% after Mn supplementation.

Concerning the impact of antifungal cultures on the growth rate of fungi, important decreases in growth rates, ranging from >20% to ~90%, were observed for 14 out of the 25 tested fungi in the *L. plantarum* L244 and *L. rhamnosus* CIRM-BIA1759 whey without Mn supplementation as compared to the control condition (Figure 5B and Appendix A). A relative growth rate reduction was observed for *P. adametzoides* (92.50% and 87.50%), *R. mucilaginosa* (68.93% and 83.50%), *C. suis* (73.02% and 77.78%), *P. solitum* (57.14% and 76.62%), *D. hansenii* (77.42% and 48.39%), *T. asahii* (54.24% and 52.54%), *C. sphaerospermum* (38.24% and 66.18%), *P. charlesii* (55% and 60%), *P. pinodella* (58.23% and 55.70%), *S. candida* (45.71% and 57.14%), *C. parapsilosis* (50% and 50%), *P. antarticum* (42.50% and 50%), *P. bialowiezense* (64.29% and 35.71%) and *M. guilliermondii* (29.31% and 31.03%) in *L. plantarum* L244 and *L. rhamnosus* CIRM-BIA1759 whey without manganese, respectively. Furthermore, *M. circinelloides*, *P. biforme* and *Y. lipolytica* also exhibited a relative growth rate reduction (35.63%, 23.08% and 20.45%, respectively), but only in the *L. plantarum* L244 whey, while *P. roqueforti* exhibited a growth rate reduction (32.32%), but only in the *L. rhamnosus* CIRM-BIA1759 whey. After Mn supplementation in *L. plantarum* L244 or *L. rhamnosus* CIRM-BIA1759 whey, almost completely or completely restored growth rates (70 to 100% according to the considered target) (Figure 5B) were observed in comparison to the control whey, with the exception of *P. roqueforti*. It thus confirmed the antifungal activity suppression or reduction by Mn supplementation, as previously observed in the yogurt model.

Interestingly, for the other fungal targets (i.e., *K. lactis*, *P. fermentans*, *C. inconspicua*, *T. elegans*, *C. intermedia*, *G. geotrichum* and *M. racemosus*), no or very little effects on growth parameters were observed for both antifungal strain wheys, indicating that Mn scavenging by antifungal LAB had little impact on their growth behavior and that overall these species were not affected by the antifungal activity of these LAB. Overall, these results based on growth parameters were in agreement with the visual observations from the oCelloScope images, with the exception of *M. circinelloides* in *L. plantarum* L244 whey and *P. roqueforti* in *L. rhamnosus* CIRM-BIA1759 whey, for which no visual inhibition was noticeable. On the contrary, while a visual inhibition of *P. biforme* in *L. rhamnosus* CIRM-BIA1759 whey was observed in oCelloScope images, this was not clearly confirmed after the growth parameter determination.

## 4. Discussion

The aim of this study was to extend the reported implications of competition exclusion for Mn in the antifungal activity of two antifungal LAB strains, *L. plantarum* L244 and *L. rhamnosus* CIRM-BIA1759, against some dairy product fungal spoilers [14]. To do so, we investigated if competitive exclusion for other trace elements (metals and vitamins) could be involved and if this bioprotective mechanism was applicable to a larger panel of fungal species. A yogurt-mimicking model, previously developed by Garnier et al. [21], was chosen in order to be as close as possible to the final products, thus avoiding potential discrepancies between the activities observed on a synthetic medium and tests in actual food products as already reported [10,28]. Firstly, we determined if these two LAB strains exhibited an antifungal activity against four fungal targets isolated from spoiled dairy products and representative of the most common dairy product spoilers [1,10]. The results obtained in the presence of the antifungal bacteria and in the absence of trace element supplementation indicated that both *L. plantarum* L244 and *L. rhamnosus* CIRM-BIA1759 exhibited antifungal activities towards *P. biforme*, while only the *L. plantarum* L244 strain inhibited *Y. lipolytica* in comparison to the controls (i.e., without antifungal bacteria, and with or without trace element addition). However, no antifungal activity was observed against *M. racemosus* and *G. geotrichum*. Leyva Salas et al. [22] found that *L. plantarum* L244 exhibited similar antifungal activities in a yogurt model against the same fungal strains as tested in the present study. However, the time to visible growth of *P. biforme* was delayed by four additional days as compared to the control, and *M. racemosus* showed an intermediate inhibition that was not observed in the present study. It should be noted that the methodological approaches were identical with the exception that yogurts were prepared from skimmed milk in the present study, while semi-skimmed milk was used by Leyva-Salas et al. [22], thus potentially affecting the metabolic processes related to antifungal activity. To study the possible involvement of competition exclusion in the observed antifungal activities, 6 metals and 12 vitamins were individually supplemented prior to milk fermentation at 10 times the reported concentrations in yogurt. These trace elements were selected based on their essential role in LAB and fungal growth [29,30], and also because of their limiting concentration in yogurt. Among the tested trace metal elements, only Mn supplementation resulted in antifungal activity suppression as observed through the restored growth of *P. biforme* and *Y. lipolytica*. This confirmed the previous results from Siedler et al. [14] and Shi and Knøchel [31,32] obtained on various fungal targets using different selected antifungal LAB, including *L. plantarum, L. rhamnosus* and *L. paracasei*. The depletion of manganese as a bioprotective strategy has been patented (“Inhibition of fungal growth by manganese depletion”, EP3780962A2). Despite the fact that it has been reported in the literature that certain LAB species and strains, including *Lactobacillus* spp., can sometimes decrease the vitamin content of fermented dairy products, such as folate [33] and riboflavin [34], none of the tested vitamin supplementations alleviated antifungal activity, suggesting that competition exclusion for the tested vitamins is not involved in *L. plantarum* L244 and *L. rhamnosus* CIRM-BIA1759 antifungal activity.

We then analyzed yogurt-derived cell-free whey, obtained after 2 weeks of yogurt storage at 10 °C, for trace metal elements using HR-ICP-MS and confirmed that Mn availability was strongly decreased by both tested antifungal LAB, as Mn concentrations were reduced over 10-fold as compared to the control (<0.5 versus 8 ng/g in the control). The ability of *L. plantarum* and *L. rhamnosus* to accumulate high intracellular levels of Mn, in particular to cope with oxidative stress, is a well-known phenomenon [15]. In agreement with the present study’s HR-ICP-MS results, Siedler et al. [14] also showed that an antifungal co-culture consisting of *L. rhamnosus* and *L. paracasei* decreased over 10-fold the Mn levels of a yogurt supernatant, with a concentration lower than the detection limit (3 ng/g) after fermentation.

It is important to note that HR-ICP-MS analyses on yogurt were solely performed after 2 weeks of storage in the present study. This endpoint for yogurt storage was chosen to be in agreement with yogurt shelf-life, which is a few weeks, and also as oCelloScope preliminary data showed that inhibitions were more noticeable after two weeks of storage compared to one week. A further investigation of the kinetics of manganese depletion in yogurt would be of great interest, as they are critical for biopreservation. Indeed, if the scavenging is linear in time or occurs quickly, it will change the efficacy of the antifungal control. From an applied point of view, a rapid depletion of Mn is desirable, as fungal spoilage can occur at different stages, from dairy-processing units to the consumer’s home. In this context, it should be noted that Siedler et al. [14] measured the Mn scavenging of a bioprotective solution (a mix of *L. rhamnosus* and *L. casei*) post-fermentation (about 6 h) and observed a 10-fold reduction in Mn content, indicating that the scavenging occurs rapidly. Moreover, it would be of interest to determine the expression of manganese transporters by both antifungal strains to evaluate if it is driven by active transport or passive binding. Siedler et al. [14] showed that the manganese transporter encoding gene (*mntH1*) was among the most expressed genes during milk fermentation by antifungal *L. rhamnosus* and *L. paracasei* (fifth- and seventh-highest read counts of all genes, respectively, accounting for up to 1.8% of all transcripts) and that these genes were still highly expressed during cold storage (400 to 500 times above median transcript levels). Interestingly, the copper concentration was also found to be significantly reduced by *L. rhamnosus* CIRM-BIA1759, but residual Cu concentrations were apparently not limiting. In *Lactobacillus*, no function for copper has been identified, but copper binding has been reported in lactobacilli, which may explain this result [35,36]. Given the fact that copper is also an essential trace metal for fungi, the selection of antifungal LAB strains with a strong copper biosorption capacity could also be of interest, especially if they exhibit other antifungal mechanisms.

We then investigated how manganese depletion by antifungal LAB affected the trace element uptake for the four initial tested fungal targets. After 7 days of incubation in antifungal cell-free whey, we observed that biomass yields were generally lower than those observed in the control whey, with a species-dependent effect quite similar to those observed in the yogurt model. More interestingly, we showed that element uptake profiles in the whole-cell biomass of all tested fungal targets were significantly affected by the use of antifungal LAB. Concerning Mn, we observed that fungal uptake was minimal in Mn-depleted whey, while it was accumulated at high concentrations in the control. This is in agreement with the results from Wehmeier et al. [26], which showed that the availability of Mn influenced its uptake by *Candida albicans*. Indeed, while 16% of the available Mn was uptaken in a poor medium, 79% were accumulated in a rich medium. We also showed that Zn uptake was significantly higher for three out of the four tested fungal species (i.e., *P. biforme*, *G. geotrichum* and *Y. lipolytica*). Given the fact that Mn is a cofactor of cytosolic and/or mitochondrial SOD in fungi [37,38,39], we can hypothesize that, under Mn starvation, Zn uptake may be upregulated to increase Cu/Zn-SOD, which also plays a role in protection against oxidative stress. In contrast, we observed a significant decrease in Ca uptake in fungi cultivated in yogurt whey produced with antifungal strains, suggesting an alteration in the homeostasis of this trace element. In fungi, several processes, including stress response, cell wall integrity and adaptation to osmotic, temperature, saline and pH variations, are controlled by Ca^2+^ signals [40]. Ca^2+^ also plays a role in the hyphal tip growth [41]. However, based on these results, it cannot be concluded whether the differences in trace element uptake by fungi after cultivation in antifungal and control whey are due to the disruption of Mn homeostasis, the presence of inhibitory metabolites produced by antifungal LAB or both. Investigating the transcriptomic and proteomic responses of fungi could enhance our understanding of the action mode of the tested antifungal LAB.

Finally, in order to determine both qualitatively and quantitatively the antifungal activity spectrum, and whether competitionexclusion for Mn was a generic antifungal mechanism among dairy spoilage fungi, we followed the growth kinetics of 25 fungal species in either *L. plantarum* L244 and *L. rhamnosus* CIRM-BIA1759 whey, and with or without post-fermentation Mn supplementation at 2.9 µg/g. Mn supplementation restored partially or totally the growth of 16 out of the 25 tested fungi that were inhibited in the antifungal whey without Mn supplementation. They corresponded, respectively, to 9 out of 13 tested molds and 7 out of 12 tested yeasts (Figure 4). *C. sphaerospermum*, *P. pinodella* and *S. candida*, as well as all, except one, *Penicillium* species tested in our study, were inhibited in both antifungal wheys, and Mn supplementation had a clear effect on their growth behavior. In addition, these mold species presented clear alterations in their morphology (i.e., swollen and irregular hyphae), which were similar to those observed in *Aspergillus niger* grown under Mn deficiency during citric acid fermentation [42]. Notably, Aunsbjerg et al. [43] also observed the same effect on *Penicillium* sp. DCS 1541 morphology with an oCelloScope in the presence of 0.5 mg/mL propionic acid, suggesting that these microscopic structural changes may also be the result of organic acid production by the antifungal LAB.

The only exception among *Penicillium* spp. was *P. roqueforti*, for which a slight inhibition was observed, but no effect of Mn supplementation, suggesting that the low Mn levels found in antifungal LAB cell-free whey had little effect on its growth. Similar results were also obtained by Siedler et al. [14] and Shi and Knøchel [31]. Another interesting finding was that none of the tested fungi from the Mucoromycota phylum (i.e., *M. racemosus*, *M. circinelloides* and *T. elegans*) were inhibited in the antifungal cell-free whey and/or yogurt model, as well as several yeasts species (*C. inconspicua*, *C. intermedia*, *G. geotrichum*, *K. lactis* and *P. fermentans*), despite the low residual levels of Mn available in antifungal cell-free whey. Based on the observations for the Mucoromycota phylum, it could be of interest in future research to further explore if a phylogenetic pattern can be observed by extending the interspecies diversity. We can hypothesize that the species mentioned above have lower Mn requirements or that they can better adapt to the stress caused by Mn scarcity, as well as that caused by inhibitory metabolites such as organic acids or volatile compounds that are produced by antifungal LAB. Indeed, the role of these molecules in antifungal activity cannot be neglected, as it is often observed that the antifungal activity of LAB is severely decreased after pH neutralization (because of their charged form, which prevents them from diffusing across the cell membrane). Moreover, it has been shown in yeasts that weak organic acids such as benzoic acid are pro-oxidant, thus increasing the effects of ROS production by the respiratory chain [44]. Interestingly, diacetyl production, which is associated with *L. plantarum* L244 antifungal activity [13], also possesses a pro-oxidant effect, which may in turn lead to damages to plasma membrane function and integrity due to the ROS oxidation of membrane lipids [45]. We can therefore speculate that, for sensitive fungal species, the inhibitory activity by antifungal LAB could result from a simultaneous stress in which antifungal molecule production increases the effects of ROS production by the respiratory chain, which in turn cannot be compensated through Mn-SOD due to the scarcity of Mn. Taken together, this may also partly explain why facultative aerobic yeast species such as *C. inconspicua*, *C. intermedia*, *P. fermentans* and *K. lactis* were not inhibited in the tested conditions. A further investigation on the Mn requirements of inhibited and non-inhibited fungi, as well as the effects of AF cell-free whey on ROS accumulation, could confirm these hypotheses. The discrepancy observed between the visual inhibition of *P. biforme* in the presence of *L. rhamnosus* CIRM-BIA1759 and the calculated growth parameters can be explained by the difficulty of modeling with QurvE in this specific case.

In conclusion, this study provided new insights about the competition exclusion mechanism for two antifungal LAB strains against a variety of fungal spoilers isolated from dairy products. It showed that competition exclusion for manganese plays a central role in the observed antifungal activity, with Mn supplementation leading to partial or total growth restoration for fungi initially inhibited by both AF LAB strains. However, we also showed that several dairy fungal spoilers were not sensitive to the antifungal activity regardless of the mechanism involved, including competitive exclusion for Mn, thus highlighting that this latter mechanism is not generic among fungi, and further research is needed to find efficient natural solutions to control these spoilers. Overall, antifungal activity is most certainly multifactorial, relying on different combined mechanisms (e.g., competitionexclusion for Mn, antifungal metabolite production of organic acids, volatile compounds, peptides) which altogether contribute to inducing a sequential stress in the susceptible fungi.

## Figures and Tables

**Figure 1 microorganisms-13-02543-f001:**
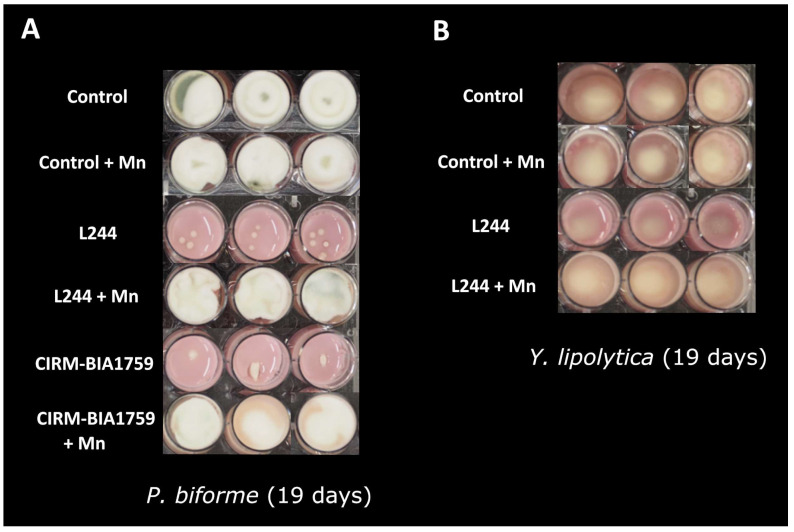
Pictures of the model yogurt produced with the MY800 starter and inoculated with *P. biforme* (**A**) and *Y. lipolytica* (**B**) after 19 days of incubation at 10 °C (nine replicates per condition). Control: yogurt without antifungal agent, L244 and CIRM-BIA1759: addition of the antifungal *L. plantarum* L244 or *L. rhamnosus* CIRM-BIA1759 strains during the fermentation, respectively; L244 + Mn and CIRM-BIA1759 + Mn: addition of the antifungal *L. plantarum* L244 or *L. rhamnosus* CIRM-BIA1759 strains during the fermentation and manganese supplementation (2.9 µg/g) in the milk, respectively.

**Figure 2 microorganisms-13-02543-f002:**
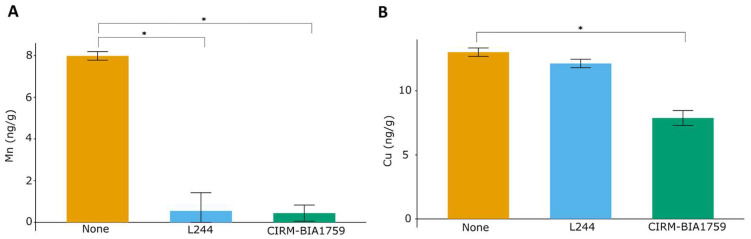
Trace metal element determined through HR-ICP-MS. Mn (**A**) and Cu (**B**) content in whey from yogurt produced either with only MY800 (None), or with addition of the antifungal *L. plantarum* L244 (L244) or *L. rhamnosus* CIRM-BIA1759 (CIRM-BIA1759) strains, after two weeks of storage at 10 °C (4 replicates/condition). One-way analysis of variance (ANOVA) was performed (*p* < 0.05). An asterisk indicates a statistically significant difference between control whey and antifungal whey concentrations according to a Student *t*-test (*p* < 0.05).

**Figure 3 microorganisms-13-02543-f003:**
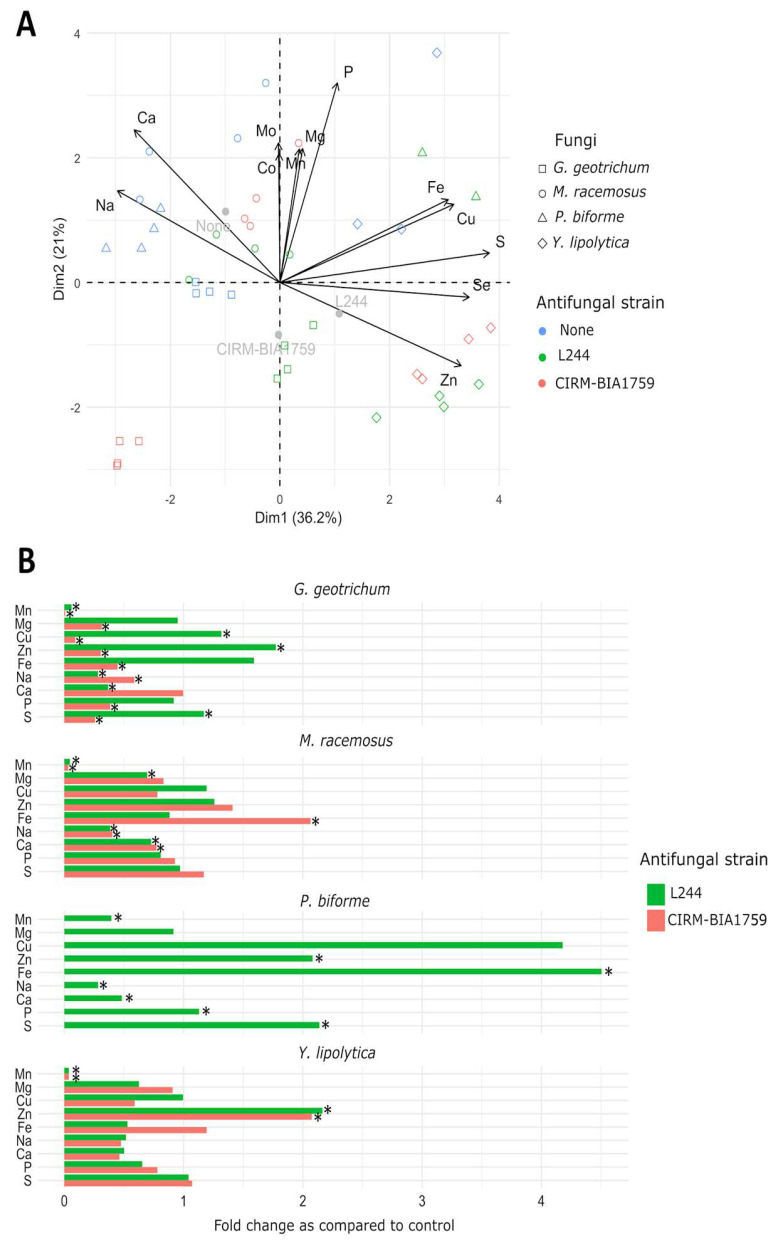
PCA (**A**) and trace element uptake fold change as compared to control (**B**), showing profile of *G. geotrichum*, *M. racemosus*, *P. biforme* and *Y. lipolytica* in whey from yogurts produced either only with MY800 (None) or with addition of either the antifungal *L. plantarum* L244 (L244) or *L. rhamnosus* CIRM-BIA1759 (CIRM-BIA1759) strains after 2 weeks of storage at 10 °C (four replicates/condition). An asterisk indicates a statistically significant difference between control whey and antifungal whey concentrations according to a Student *t*-test (*p* < 0.05).

**Figure 4 microorganisms-13-02543-f004:**
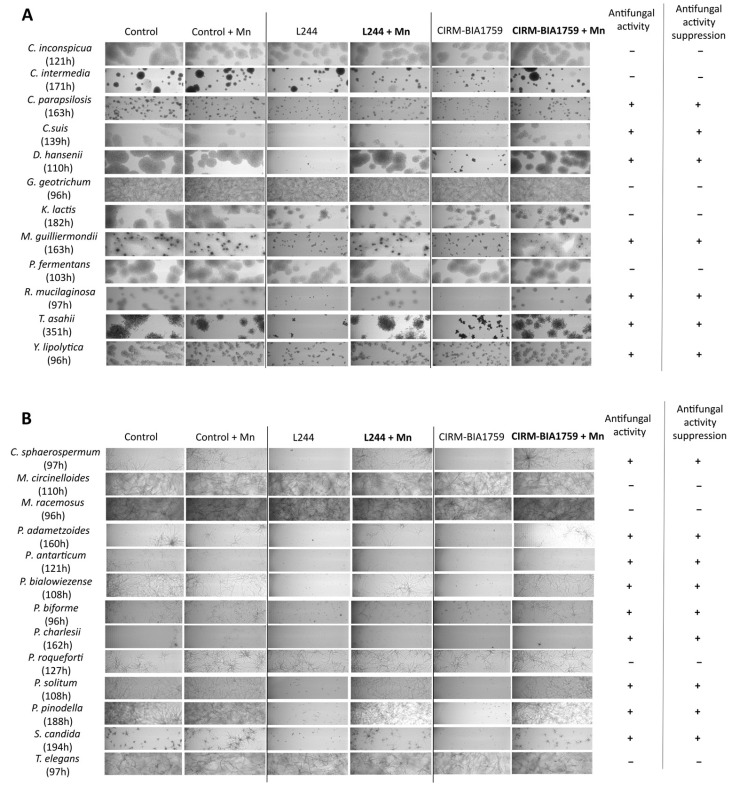
oCelloScope pictures of yeast (**A**) and mold (**B**) growth at a given time in whey produced either with only the MY800 starter (control) or *L. plantarum* L244 or *L. rhamnosus* CIRM-BIA1759 supplemented or not supplemented with manganese (2.9 µg/g). All experiments were performed in triplicates.

**Figure 5 microorganisms-13-02543-f005:**
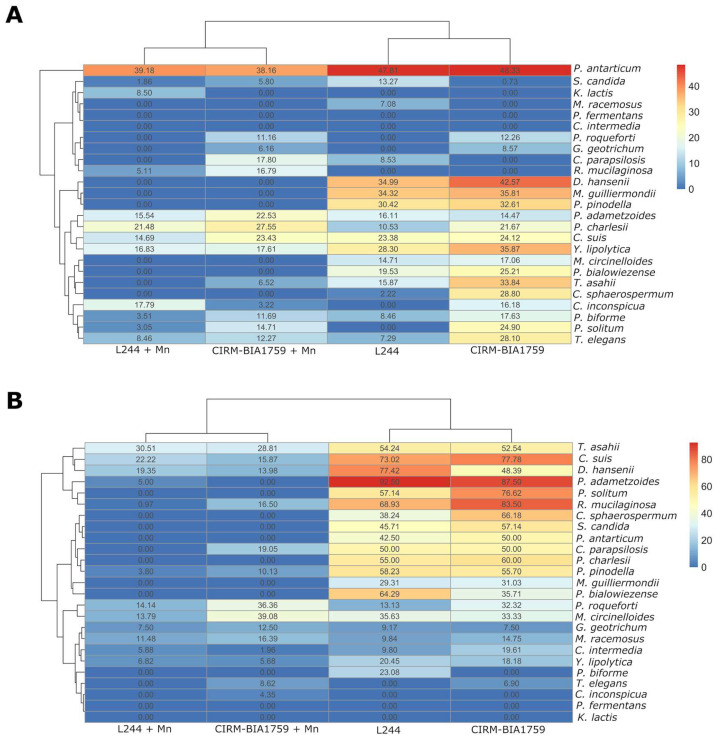
Heatmap representing growth parameters (percentages of (**A**) increase in lag time and (**B**) decrease in relative µmax) of the 25 fungal targets tested in whey produced with *L. plantarum* L244 or *L. rhamnosus* CIRM-BIA1759 and supplemented or not supplemented with Mn at 2.9 µg/g (three replicates/condition).

## Data Availability

The original contributions presented in this study are included in the article/Appendix A; further inquiries can be directed to the corresponding author.

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
