# Peer review of "Competition-Exclusion for Manganese Is Involved in Antifungal Activity of Two Lactic Acid Bacteria Against Various Dairy Spoilage Fungi"

_microorganisms, 2025, doi:10.3390/microorganisms13112543_

Round 1

Reviewer 1 Report

Comments and Suggestions for Authors

The use of lactic acid bacteria in biopreservation has become a highly topical research area, representing a viable alternative to the use of chemical preservatives and, at the same time, a complementary tool for preventing fungal spoilage of dairy products. Within the antifungal mechanisms attributed to lactic acid bacteria, the competition-exclusion process for oligoelements (trace mineral elements) has recently been identified as having a significant role in inhibiting the growth of spoilage fungi. To further investigate this mechanism, two strains of lactic acid bacteria with antifungal activity — Lactiplantibacillus plantarum L244 and Lactobacillus rhamnosus CIRM-BIA1759 — were investigated using an experimental yogurt model. The study evaluated the antifungal capacity of these strains against four fungal species frequently involved in spoilage of dairy products: Penicillium biforme, Mucor racemosus, Galactomyces geotrichum, and Yarrowia lipolytica. The tests were performed both under standard conditions and with controlled supplementation of trace elements (six metals and twelve vitamins). The results revealed that only manganese (Mn) supplementation suppressed the antifungal activity of both bacterial strains against P. biforme and Y. lipolytica species, suggesting a direct dependence between manganese availability and antifungal efficiency of these bacteria. To confirm the hypothesis of competition for trace elements, the metal sequestration capacity was analyzed by high-resolution mass spectrometry (HR-ICP-MS) in both cell-free yogurt whey and fungal biomass. The results indicated a significant uptake of manganese by both bacterial strains, as well as copper (Cu) in the case of L. rhamnosus CIRM-BIA1759 strain. At the same time, it was found that the absorption profile of mineral and non-metallic elements of the fungi was significantly modified by the presence of these bacterial cultures, which confirms an indirect metabolic impact on fungal development. In a subsequent stage, the influence of manganese competition on the inhibition of a larger number of spoilage fungi (25 species) was evaluated using the oCelloScope system for dynamic growth monitoring. Supplementation of the medium with manganese eliminated the antifungal effect of the LAB strains in 16 of the 25 cases tested, which confirms that the inhibition effect is dependent on the limitation of manganese in the medium, but is not universally valid for all fungal species. In conclusion, the study demonstrates that the strains Lactiplantibacillus plantarum L244 and Lactobacillus rhamnosus CIRM-BIA1759 exert an antifungal activity partially conditioned by their ability to compete with fungi for manganese, and this biopreservation mechanism, although effective for certain species, requires a differentiated approach depending on the characteristics of each target microorganism. This finding reinforces the concept that biopreservation represents a complex ecological strategy, based on subtle trophic and metabolic interactions between microorganisms, with a high potential for applicability in the modern food industry.

Author Response

Thank you for your analysis of our article.

Reviewer 2 Report

Comments and Suggestions for Authors

General assessment

The manuscript investigates the antifungal effects of Lactiplantibacillus plantarum L244 and Lacticaseibacillus rhamnosus CIRM-BIA1759 in a yogurt model, with a focus on manganese (Mn) depletion as the main inhibitory mechanism. The study combines yogurt fermentation, trace element supplementation, HR-ICP-MS quantification of whey and fungal biomass, and oCelloScope-based growth kinetics across 25 fungal species. Supplementary data provide lag times, growth rates, and detailed elemental profiles. The present study builds upon prior work by Siedler et al. (2020), who showed that Mn depletion was a key antifungal mechanism in LAB, and extends this concept by evaluating its specificity across 25 fungal species. Furthermore, the core concept of Mn depletion as a strategy for fungal inhibition is also covered by the international patent “Inhibition of fungal growth by manganese depletion” (AU2019254580A1 / WO2019202003A2 / US11707070B2). This patent describes the use of Mn depletion by LAB—including L. plantarum and L. rhamnosus—as a biopreservative approach in food matrices. While the manuscript does not appear to duplicate the patented claims, it clearly builds upon them.

The manuscript is well designed and clearly written, addressing an important applied problem in dairy spoilage and biopreservation. With appropriate revisions to acknowledge the overlap with existing intellectual property and to strengthen the mechanistic discussion, the study should be suitable for publication. Please see the comments below.

Major comments

  1. For transparency and to delineate the novelty of the current contribution, the authors should explicitly acknowledge the patent and clarify how their findings extend, validate, or differentiate from the prior art. To my opinion the main contribution of this manuscript is in its scope, detail, and mechanistic insights, rather than in introducing new concepts.
  2. The observation that 9 fungi were not inhibited is interesting. The discussion offers good hypotheses (lower Mn requirements, better stress adaptation), but this could be explored slightly further. Is there any phylogenetic pattern? The Mucoromycota phylum members (M. racemosus, M. circinelloides, T. elegans) were all resistant. Could this be a taxonomic trait worth highlighting as a point for future research?
  3. The HR-ICP-MS measurements were taken only at a late single time point (1–2 weeks). Since the kinetics of Mn depletion are critical for biopreservation, this limitation should be acknowledged and discussed.
  4. The manuscript concludes that Mn competition is central but not the sole mechanism. The discussion on the potential synergy with organic acids/diacetyl is very good. It would strengthen the conclusion if the authors explicitly stated that the antifungal activity likely represents a sequential stress effect, with Mn competition as a major, but not the only, contributing factor.
  5. The manuscript could benefit from additional clarification regarding the mechanistic basis of Mn depletion, specifically, whether it is driven by active transport (e.g., mntH1 expression) or passive binding.

Minor comments

  1. Some figure and table legends lack details on sample size (n), statistical tests, and units—please make legends fully self-contained.
  2. Ensure consistent units for elemental data (e.g., ng/g vs. mg/L) across figures and text.
  3. Correct typographical errors such as “after after” in supplementary figure captions.
  4. Clarify abbreviations (e.g., HR-ICP-MS, µ) at first mention in the main text and figures.
  5. Ensure references are formatted consistently.
  6. Review the manuscript for minor grammatical edits (articles, punctuation, and consistent tense).

Author Response

Thank you for your constructive comments.

Major comments

  1. For transparency and to delineate the novelty of the current contribution, the authors should explicitly acknowledge the patent and clarify how their findings extend, validate, or differentiate from the prior art. To my opinion the main contribution of this manuscript is in its scope, detail, and mechanistic insights, rather than in introducing new concepts.

We agree. The patent was acknowledged in the introduction (lines 76-78) and a precision on how our study extends from it was added (lines 588-592). We hope this is satisfactory for you.

  1. The observation that 9 fungi were not inhibited is interesting. The discussion offers good hypotheses (lower Mn requirements, better stress adaptation), but this could be explored slightly further. Is there any phylogenetic pattern? The Mucoromycota phylum members (M. racemosus, M. circinelloides, T. elegans) were all resistant. Could this be a taxonomic trait worth highlighting as a point for future research?

Thank you for this comment. We added a comment on this in the discussion (lines 557-559).

  1. The HR-ICP-MS measurements were taken only at a late single time point (1–2 weeks). Since the kinetics of Mn depletion are critical for biopreservation, this limitation should be acknowledged and discussed.

 Indeed, depletion kinetics can have a considerable impact on the activity. This limitation was already mentioned and discussed in the original manuscript but we extended this point (L481-493).

  1. The manuscript concludes that Mn competition is central but not the sole mechanism. The discussion on the potential synergy with organic acids/diacetyl is very good. It would strengthen the conclusion if the authors explicitly stated that the antifungal activity likely represents a sequential stress effect, with Mn competition as a major, but not the only, contributing factor.

Thank you for your proposal. This statement was added in the conclusion (lines 592-596).

  1. The manuscript could benefit from additional clarification regarding the mechanistic basis of Mn depletion, specifically, whether it is driven by active transport (e.g., mntH1 expression) or passive binding.

An additional clarification about manganese transport was implemented (line 76). We hope this is satisfactory for you.

Minor comments

  1. Some figure and table legends lack details on sample size (n), statistical tests, and units—please make legends fully self-contained.

Legends were completed accordingly. Moreover, figure 3 was improved, with statistical tests implemented in figure 3B, and figure S2 was improved to be homogeneous in comparison to the other figures.

  1. Ensure consistent units for elemental data (e.g., ng/g vs. mg/L) across figures and text.

Units were homogenized accordingly.

  1. Correct typographical errors such as “after after” in supplementary figure captions.

The typographical errors were corrected .

  1. Clarify abbreviations (e.g., HR-ICP-MS, µ) at first mention in the main text and figures.

The abbreviations were clarified at first mention in the main text and figures.

  1. Ensure references are formatted consistently.

The references were reformatted.

  1. Review the manuscript for minor grammatical edits (articles, punctuation, and consistent tense).

The manuscript was proofread and minor grammatical edits were implemented.

Reviewer 3 Report

Comments and Suggestions for Authors

In this manuscript, the authors examined metals and vitamins in antifungal activity of lactic acid bacteria. My comments are as follows:

  1. Page 1, line 35: Bacteria name should be italic.

  1. Page 3, line 134: 107 CFU/mL. I wondered if 7 should be superscript?

  1. Page 4, lines 140-142: Vitamin B8 is inositol which is not included in vitamin B complex. In this study, the other seven vitamin B are undoubtedly included in B complex. The authors should explain how come you did not choose vitamin B7 (biotin) instead of using B8.

  1. Page 8: the authors can bring Figure 3 from page 9 to this page.

  1. Page 10, line 344: C. suis might change place with C. parapsilosis.

  1. Page 11, lines 352-359: In this paragraph, bacteria name are not italic. Please correct it.

  1. Page 13: in line 436, Garnier et al., (2018) [21]; in line 446, Leyva Salas et al. [22]. Please be consistent!

  1. Page 17: Please extend ‘Abbreviations.’

Author Response

 Thank you for your constructive comments.

  1. Page 1, line 35: Bacteria name should be italic.

  The text was modified accordingly (lines 54-55).

  1. Page 3, line 134: 107 CFU/mL. I wondered if 7 should be superscript?

  Indeed, 7 must be superscript, the text was modified accordingly (line 144)

  1. Page 4, lines 140-142: Vitamin B8 is inositol which is not included in vitamin B complex. In this study, the other seven vitamin B are undoubtedly included in B complex. The authors should explain how come you did not choose vitamin B7 (biotin) instead of using B8.

  Thank you, biotin was actually tested in this study and was erroneously noted as B8 vitamin, the mistake was corrected (line 150) and modified in table S1 (supplementary data).

  1. Page 8: the authors can bring Figure 3 from page 9 to this page.

  The modification was done (page 8).

  1. Page 10, line 344: C. suis might change place with C. parapsilosis.

  The modification was done (line 347).

  1. Page 11, lines 352-359: In this paragraph, bacteria name are not italic. Please correct it.

  The modifications were done (lines 355-362).

  1. Page 13: in line 436, Garnier et al., (2018) [21]; in line 446, Leyva Salas et al. [22]. Please be consistent! 

 The references were homogenized.

  1. Page 17: Please extend ‘Abbreviations.’

 The abbreviations were extended (line 648).

Reviewer 4 Report

Comments and Suggestions for Authors

To maximise the scientific impact and address the identified limitations, the following improvements and supplementary experiments are recommended by me:

  • HR-ICP-MS analysis was performed only after a 2-week storage period. The kinetics of Mn depletion in yoghurt are not reported, which is crucial from an applied perspective for preventing rapid spoilage. The existing data from HR-ICP-MS already show significant changes in fungal element uptake profiles when grown in manganese (Mn)-depleted whey. The discussion should be significantly expanded to leverage these specific, quantitative findings.
  • The manuscript currently notes that it is difficult to conclude whether inhibition results from Mn disruption, inhibitory metabolites, or both. Since new experiments to separate these effects are impossible, the authors must integrate the existing data with the literature to present a cohesive synergistic model as the most likely mechanism.
  • The authors must address the noted discrepancy where visual inhibition of P. biforme was observed in L. rhamnosus CIRM-BIA1759 whey images, but the calculated growth parameters did not clearly confirm this. The revision should re-examine the quantitative data and, if robust, discuss why visual observation might be misleading for quantitative analysis. Or, if the visual observation is deemed more representative of long-term spoilage, justify the text for relying more heavily on the visual data in this specific case, or adjust the interpretation to reflect only partial inhibition.
  • Since the study relies on data collected at a single time point (2 weeks of storage for whey analysis), this limitation must be transparently addressed. The authors should explicitly state in the Methods or Discussion that HR-ICP-MS was performed after 2 weeks of storage and discuss why this endpoint is relevant for characterising Mn availability during typical cold storage of yoghurt.
  • The finding that 9 out of 25 fungi were not or poorly inhibited, despite Mn depletion, is a critical novel insight. Reiterate that Mn scavenging is not a generic antifungal mechanism among all fungal species. Discuss potential explanations for non-inhibition - these species may possess inherently lower Mn requirements or better adaptive mechanisms to overcome Mn scarcity and metabolite stress.

Author Response

Thank you for your constructive comments.

  • HR-ICP-MS analysis was performed only after a 2-week storage period. The kinetics of Mn depletion in yoghurt are not reported, which is crucial from an applied perspective for preventing rapid spoilage. The existing data from HR-ICP-MS already show significant changes in fungal element uptake profiles when grown in manganese (Mn)-depleted whey. The discussion should be significantly expanded to leverage these specific, quantitative findings.

 We agree, and this was addressed in part by reviewer 2. These aspects were expanded in the discussion (L481-493 And L513-517).

  • The manuscript currently notes that it is difficult to conclude whether inhibition results from Mn disruption, inhibitory metabolites, or both. Since new experiments to separate these effects are impossible, the authors must integrate the existing data with the literature to present a cohesive synergistic model as the most likely mechanism.

 An additional note has been added to conclude on the multifactorial aspect of antifungal activity (lines 570-573). We hope this is satisfactory for you.

  • The authors must address the noted discrepancy where visual inhibition of P. biforme was observed in L. rhamnosus CIRM-BIA1759 whey images, but the calculated growth parameters did not clearly confirm this. The revision should re-examine the quantitative data and, if robust, discuss why visual observation might be misleading for quantitative analysis. Or, if the visual observation is deemed more representative of long-term spoilage, justify the text for relying more heavily on the visual data in this specific case, or adjust the interpretation to reflect only partial inhibition.

              The discrepancy observed was addressed in lines 580-583. Indeed, in this specific case, visual observation was more representative and the modelisation of growth parameters was difficult.

  • Since the study relies on data collected at a single time point (2 weeks of storage for whey analysis), this limitation must be transparently addressed. The authors should explicitly state in the Methods or Discussion that HR-ICP-MS was performed after 2 weeks of storage and discuss why this endpoint is relevant for characterising Mn availability during typical cold storage of yoghurt.

We agree and this limitation was addressed lines 481-482. A supplementary explanation to discuss why this endpoint of two-week storage is relevant was added lines 482-483.

  • The finding that 9 out of 25 fungi were not or poorly inhibited, despite Mn depletion, is a critical novel insight. Reiterate that Mn scavenging is not a generic antifungal mechanism among all fungal species. Discuss potential explanations for non-inhibition - these species may possess inherently lower Mn requirements or better adaptive mechanisms to overcome Mn scarcity and metabolite stress.

 A mention stating that competition-exclusion is not generic among fungi was emphasized in the abstract and added line 34. The potential explanations for non-inhibition was already discussed lines 559-562 “We can hypothesize that these species have lower Mn requirements or that they can better adapt to the stress caused by Mn scarcity as well as that caused by inhibitory metabolites such as organic acids or volatile compounds that are produced by antifungal LAB”. We hope this is satisfactory.

Reviewer 5 Report

Comments and Suggestions for Authors

In this paper, the authors provide new insights about the competition-exclusion mechanism for two antifungal LAB strains against a variety of fungal spoilers isolated from dairy products.

 The article's content complies with the scope of the journal Microorganisms and may be accepted for publication.

There are some minor issues.

The Latin names of bacteria should be italicized everywhere, including keywords, Lines 357-358.

Line 134? 159: “t 107 CFU/mL” should it be  107 CFU/mL?

Line 143 “4h” number should be separated from the Unit by the space.

Line 179: “cells/ml in a 100-ml Erlenmeyer flask containing 30 ml “ ml or mL?

Line 257, 279 etc. Use the same letter (p or P) over the text to show statistical difference.

Author Response

Thank you for your constructive comments

1.The Latin names of bacteria should be italicized everywhere, including keywords, Lines 357-358.

This was amended throughout the text.

2. Line 134? 159: “t 107 CFU/mL” should it be  107 CFU/mL?

This was amended in the text L169

3. Line 143 “4h” number should be separated from the Unit by the space.

This was amended in the text L153

4. Line 179: “cells/ml in a 100-ml Erlenmeyer flask containing 30 ml “ ml or mL?

This was amended in the text L190

5. Line 257, 279 etc. Use the same letter (p or P) over the text to show statistical difference.

This was amended throughout the manuscript.

Round 2

Reviewer 2 Report

Comments and Suggestions for Authors

Thank you for submitting the revised version of your manuscript. I have reviewed the changes carefully and I am pleased to see that you have made all the appropriate corrections and addressed the reviewers’ comments thoroughly. Your efforts have substantially improved the paper, and I am happy to recommend it for acceptance.

Reviewer 3 Report

Comments and Suggestions for Authors

I have no further questions.

Reviewer 4 Report

Comments and Suggestions for Authors

The manuscript has been sufficiently revised to warrant its publication. The authors have addressed all key concerns comprehensively and analytically.